# Should Laparoscopic Complete Mesocolic Excision Be Offered to Elderly Patients to Treat Right-Sided Colon Cancer?

Michele Mazzola [1,*], Lorenzo Ripamonti [2], Alessandro Giani [1], Pietro Carnevali [1], Matteo Origi [1], BrunocDomenico Alampi [1], Irene Giusti [1], Pietro Achilli [1], Camillo Leonardo Bertoglio [3], Carmelo Magistro [4] and Giovanni Ferrari [1]

[1]  Division of Minimally-Invasive Surgical Oncology, ASST Grande Ospedale Metropolitano Niguarda, 20162 Milano, Italy
[2]  Department of General Surgery, IRCCS san Gerardo dei Tintori, 20900 Monza, Italy
[3]  Division of General Surgery, ASST Ovest Milanese, Hospital of Magenta, 20013 Magenta, Italy
[4]  Division of General Surgery, ASST Melegnano e Martesana, Hospital of Vizzolo Predabissi, 20070 Vizzolo Predabissi, Italy
*   Correspondence: michele.mazzola@ospedaleniguarda.it; Tel.: +39-0264447918

**Abstract:** Background: Despite its potential oncologic benefit, complete mesocolic excision (CME) has rarely been offered to elderly patients. The present study evaluated the effect of age on postoperative outcomes among patients undergoing laparoscopic right colectomies with CME for right-sided colon cancer (RCC). Methods: Data of patients undergoing laparoscopic right colectomies with CME for RCC between 2015 and 2018 were retrospectively analyzed. Selected patients were divided into two groups: the under-80 group and the over-80 group. Surgical, pathological, and oncological outcomes among the groups were compared. Results: A total of 130 patients were selected (95 in the under-80 group and 35 in the over-80 group). No difference was found between the groups in terms of postoperative outcomes, except for median length of stay and adjuvant chemotherapy received, which were in favor of the under-80 group (5 vs. 8 days, $p < 0.001$ and 26.3% vs. 2.9%, $p = 0.003$, respectively). No difference between the groups was found regarding overall survival and disease free survival. Using multivariate analysis, only the ASA score > 2 ($p = 0.01$) was an independent predictor of overall complications. Conclusions: laparoscopic right colectomy with CME for RCC was safely performed in elderly patients ensuring similar oncological outcomes compared to younger patients.

**Keywords:** complete mesocolic excision; laparoscopic surgery; right colectomy





## 1. Introduction

In 2009, Hohenberger first introduced the concept of complete mesocolic excision (CME) [1], in the field of colonic surgery. CME consists of the surgical dissection along the embryological fascial planes around the mesentery, with the intact removal of the mesocolon and its lymph nodes. The other two pillars of CME are central vascular ligation, which allows the removal of the intermedium and main lymph nodes (vertical direction of lymphatic drainage), and an appropriate visceral resection margin from cancer, with the removal of pericolic lymph nodes (longitudinal direction of lymphatic drainage) [2]. According to current evidence, CME can offer a 10% additional advantage in terms of 4-year disease-free survival (DFS) rates in patients undergoing right colectomy for right colon cancer (RCC), compared to right colectomy only [3,4]. However, some authors considered CME a more complex procedure than the traditional colectomy, potentially leading to higher postoperative morbidity [5]. Due to its presumed technical difficulty, risk, and complexity, CME has rarely been offered to elderly patients with RCC [6].

Improvements in both living conditions and clinical care have resulted in relevant demographic changes withan increased number of elderly patients, especially octogenarians.

The process of aging has led the population of over 85-year-old to quadruple, with nearly a 50% increase in the annual cancer incidence [7], over the last 20 years. Colorectal cancer is one of the most common malignancies in both Western countries and Northeast Asia [8]. As a result, an increased number of elderly patients will need surgical treatment for colorectal cancer [8,9], which will force the surgical community to address the question of what is the best treatment for these patients, balancing between radical oncologic surgery, potentially broadened by more complications, and traditional surgery, with presumed worse oncologic outcomes. Although the feasibility of radical intent surgical oncology in elderly patients was reported in several fields, such as gynecologic [10], urologic [10,11], colorectal [12], liver [13,14], and pulmonary surgery [15], only a few studies evaluated the safety and oncological impact of CME on these patients [6,8].

In this study, we evaluated the effect of age on postoperative outcomes among patients undergoing laparoscopic right colectomy with CME for RCC. In particular, we aimed to compare the short-term perioperative outcomes as well as long-term oncological outcomes between elderly and younger patients.

## 2. Material and Methods

### 2.1. Study Overview

This study retrospectively evaluated data from a consecutive series of adult patients (aged ≥ 18 years), who underwent laparoscopic right colectomy with CME for RCC between 2015 and 2018 at the division of Minimally Invasive and Oncologic Surgery of Niguarda Hospital in Milan, Italy.

The exclusion criteria from the study were: (1) laparotomic resections; (2) stage IV tumors; (3) emergency surgery; (4) multivisceral resections. Patients fulfilling the inclusion criteria were included in the study and divided into two groups according to age: the under-80 group (patients under the age of 80 years) and the over-80 group (patients aged ≥ 80 years). For each group, patients baseline characteristics, surgical, pathological, and oncological outcomes were evaluated.

All gathered data were prospectively recorded in an electronic spreadsheet and anonymized prior to the analysis. This study complied with the standards of the Declaration of Helsinki and current ethical guidelines; no ethical approval was necessary owing to the retrospective, observational, and anonymous nature of this study.

### 2.2. Study Endpoints

The primary endpoint was to compare the under-80 group and the over-80 group in terms of postoperative complications. The secondary endpoints were conversion rate, R0 rate, number of lymph nodes harvested, 90-day mortality, readmission rate, overall survival (OS), and DFS.

### 2.3. Variables and Definitions

Preoperative examinations, including contrast-enhanced computer tomography and colonoscopy, were performed routinely. Serum carcinoembryonic antigen was dosed.

RCC was defined as any primary adenocarcinoma of the cecum, the ascending colon, the hepatic flexure, and the first third of the transverse colon.

In our department, CME was implemented as the standard surgical approach for RCC since September 2015.

Due to recent literature suggesting better oncologic outcomes than conventional colon resection [3], in particular for patients with stage I or II diseases, in the study period, all patients with RCC were candidates for a CME right colectomy and were offered a laparoscopic approach, with the exemption of those presenting anesthesiologic contraindications to pneumoperitoneum. All patients underwent the same surgical technique as previously described [16,17]. All procedures were performed by specialized colorectal surgeons who had already performed an average of 80 laparoscopic colorectal resections and at least 50 conventional laparoscopic right colectomies each [18].

The baseline characteristics of the patients included their age, gender, body mass index (BMI), American Society of Anesthesiologists (ASA) score, age-adjusted Charlson comorbidity index (CCI) [19], and tumor pathological features.

Surgical outcomes were the length of hospital stay (LOS), defined as the number of nights spent in the hospital from the day of the surgical procedure until discharge, overall postoperative complications, recorded at 90 days and graded according to the Clavien–Dindo classification [20], readmission, and need for a blood transfusion. Conversion from laparoscopy to open was defined as the need to complete the resection or reconstruction phase by any type of laparotomy. The pathological radicality of resection, pTNM, and pathologic stages were classified according to the 8th edition of the American Joint Committee on Cancer (AJCC) staging system [21]. Resection margin status was considered R1 when the distance between the tumor and any resection margins was $\leq 1$ mm.

OS, defined as the number of months from surgical intervention to patient death via any causes, and DFS, defined as the number of months from surgical intervention to first diagnosis of cancer recurrence, were considered as oncological outcomes. Cancer recurrence was diagnosed on the basis of clinical, radiological, and laboratory exams after local multidisciplinary tumor board discussions; histological confirmation was indicated when a clear diagnosis could not be obtained from the aforementioned techniques. Indication to adjuvant chemotherapy was provided on the basis of tumor stage, according to national and international guidelines.

*2.4. Statistical Analysis*

Data were expressed as the median and interquartile range (IQR) and number and relative percentage. Normal distribution of continuous variables was assessed with the Shapiro–Wilk test. Univariate analysis was performed, and continuous variables were analyzed using the Student's *t*-test or Mann–Whitney test, while categorical variables using Fisher's exact test or Chi-Square test, as appropriate. OS and DFS were estimated using the Kaplan–Meier method and compared using the Log-rank test between the groups.

Multivariable logistic regression was carried out to identify variables independently associated with complications after surgery: variables with $p < 0.1$ at univariate analysis were included in the multivariable model, using Firth's correction for rare events. All statistical tests were two-sided and *p* values $< 0.05$ were considered significant. Data analysis was performed using JMP®, version 16 (SAS Institute Inc., Cary, NC, USA).

**3. Results**

In the study period, a total of 130 patients underwent laparoscopic right colectomy with CME for RCC and met the inclusion criteria. From these, 95 comprised the under-80 group and 35 the over-80 group.

As shown in Table 1, at the baseline the groups significantly differed only in terms of CCI and ASA score, while no differences were found considering tumor site, tumor size, and TNM staging. No difference was found between the groups in terms of surgical outcomes (Table 2), except for median LOS, which was shorter in the under-80 group (5 days [4–7] vs. 8 days [6–10], $p \leq 0.001$). Three patients were converted to open surgery, 1 in the under-80 group and 2 in the over-80 (2.0% vs. 3.1%, $p = 1$) for adhesions (2 cases), and bleeding (1 case). The postoperative transfusion rate was 10.53% in the under-80 and 14.29% in the over-80 group, $p = 0.552$. The groups did not differ in either overall or severe complication rates (23.2% vs. 28.6%, $p = 0.525$ and 10.5% vs. 8.6%, $p = 0.742$, in the under-80 and in the over-80 group, respectively). Median CCI was 0 in both groups ($p = 0.742$). No patients died within 90 days from surgery.

**Table 1.** Clinicopathological characteristics of patients.

| | Under-80 (*n* = 95) | Over-80 (*n* = 35) | *p*-Value |
|---|---|---|---|
| Male | 50 (52) | 21 (60) | 0.454 |
| BMI | 26 (23–29) | 25 (23.8–28) | 0.545 |
| BMI > 30 | 75 (78.9) | 30 (85.7) | 0.385 |
| Charlson Comorbidity Index | 5 (4–6) | 6 (6–7) | <0.001 |
| Charlson Comorbidity Index > 6 | 80 (84.2) | 18 (51.4) | <0.001 |
| ASA | | | <0.001 |
| 1 | 19 (20) | 0 (0) | |
| 2 | 46 (48.4) | 18 (51.4) | |
| 3 | 30 (31.5) | 17 (48.5) | |
| ASA > 2 | 30 (31.6) | 17 (48.6) | 0.074 |
| CEA | 2.8 (1.3–6.9) | 4.45 (2.7–9.9) | 0.054 |
| Tumor Site | | | 0.209 |
| Cecum/Ascending colon | 75 (78.9) | 31 (88.6) | |
| Hepatic Flexure/Transverse colon | 20 (21.1) | 4 (11.4) | |
| Tumor Size | 4.1 (3–6) | 4 (2.6–5) | 0.478 |
| Grading | | | 0.932 |
| 1 | 16 (16.8) | 5 (14.3) | |
| 2 | 65 (68.4) | 25 (71.4) | |
| 3 | 14 (14.7) | 5 (14.3) | |
| T | | | 0.732 |
| 1 | 20 (21.1) | 9 (25.7) | |
| 2 | 21 (22.1) | 10 (28.6) | |
| 3 | 47 (49.5) | 14 (40) | |
| 4 | 7 (7.4) | 2 (5.7) | |
| N | | | |
| X | 1 (1.1) | 0 (0) | |
| 0 | 68 (71.6) | 29 (82.8) | |
| 1 | 18 (18.9) | 3 (8.6) | |
| 2 | 8 (8.4) | 3 (8.6) | |
| Tumor Stage | | | 0.436 |
| 1 | 37 (38.9) | 17 (48.6) | |
| 2 | 32 (33.7) | 12 (34.3) | |
| 3 | 26 (27.4) | 6 (17.1) | |
| Tumor Stage > 2 | 26 (27.4) | 6 (17.1) | 0.229 |

Data are summarized as number (%) and median (interquartile range).

All patients received an R0 resection. The median number of harvested lymph nodes was 22.5 (16–29) in the under-80 group and 18.5 (14–26) in the over-80 group (*p* = 0.267). Surgical outcomes for patients are shown in Table 2. Among the 31 patients who received an indication to adjuvant chemotherapy, 25 out of 26 (96.1%) underwent it in the under-80 group, while only 1 out of 6 (16.7%) patients in the over-80 group (*p* = 0.003). The oncological outcomes of patients are shown in Table 3. The median follow-up was 51 (IQR 44.5–61), 52 (IQR 44.5–62), and 50 (IQR 39–58) months in the overall cohort, under-80 group and over-80 group, respectively (*p* = 0.242). The overall mortality rate was 20.7% and 32.4% in the under-80 and over-80 groups, respectively (*p* = 0.183). The OS at 12, 36, and 60 months was 95.8%, 88.4%, and 81.0%, respectively, in the under-80 group and 97.1%, 77.1%,

and 61.2%, respectively, in the over-80 group ($p = 0.145$). The OS is illustrated in Figure 1. A relapse occurred in 12 (12.6%) patients in the under-80 group and in 3 (8.6%) patients in the over-80 group ($p = 0.509$). The DFS at 12, 36, and 60 months was 96.8%, 88.4%, and 88.4%, respectively, in the under-80 and 100.0%, 94.3%, and 90.2%, respectively, in the over-80 group ($p = 0.536$). The DFS is depicted in Figure 2.

**Table 2.** Surgical outcomes of patients.

| | Under-80 (*n* = 95) | Over-80 (*n* = 35) | *p*-Value |
|---|---|---|---|
| LOS, Median | 5 (4–7) | 8 (6–10) | <0.001 |
| Overall complications | 22 (23.2) | 10 (28.6) | 0.525 |
| Clavien–Dindo | | | 0.794 |
| 0 | 75 (78.9) | 27 (77.1) | |
| 1 | 2 (2.1) | 0 (0) | |
| 2 | 8 (8.4) | 5 (14.3) | |
| 3a | 3 (3.1) | 2 (5.7) | |
| 3b | 5 (5.3) | 1 (2.9) | |
| 4a | 1 (1.1) | 0 (0) | |
| 4b | 1 (1.1) | 0 (0) | |
| Severe Complication | 10 (10.5) | 3 (8.6) | 0.742 |
| Comprehensive Complication Index | 0 (0–0) | 0 (0–0) | 0.723 |
| 90-day mortality | 0 (0) | 0 (0) | 1 |
| Blood Transfusion | 10 (10.5) | 5 (14.3) | 0.552 |
| Readmission | 0 (0) | 0 (0) | 1 |
| Harvested Lymph nodes | 22.5 (16–29) | 18.5 (13.7–26) | 0.267 |
| R0 | 95 (100) | 35 (100) | 1 |
| Conversion rate | 2 (2) | 1 (3.1) | 0.8 |

Data are summarized as number (%) and median (interquartile range). Abbreviation: LOS: length of stay.

**Table 3.** Oncological outcomes of patients.

| | Under-80 (*n* = 95) | Over-80 (*n* = 35) | *p*-Value |
|---|---|---|---|
| Adjuvant chemotherapy * | 25/26 (96.1) | 1/6 (16.7) | <0.001 |
| Relapse | 12 (12.6) | 3 (8.6) | 0.509 |
| Survival rate | 65 (79.3) | 23 (67.7) | 0.183 |
| Follow-up | 43 (32.5–54.5) | 32.5 (21.8–42.3) | 0.004 |

Data are summarized as number (%) and median (interquartile range). * Calculated on 32 patients with a pathological stage > 2.

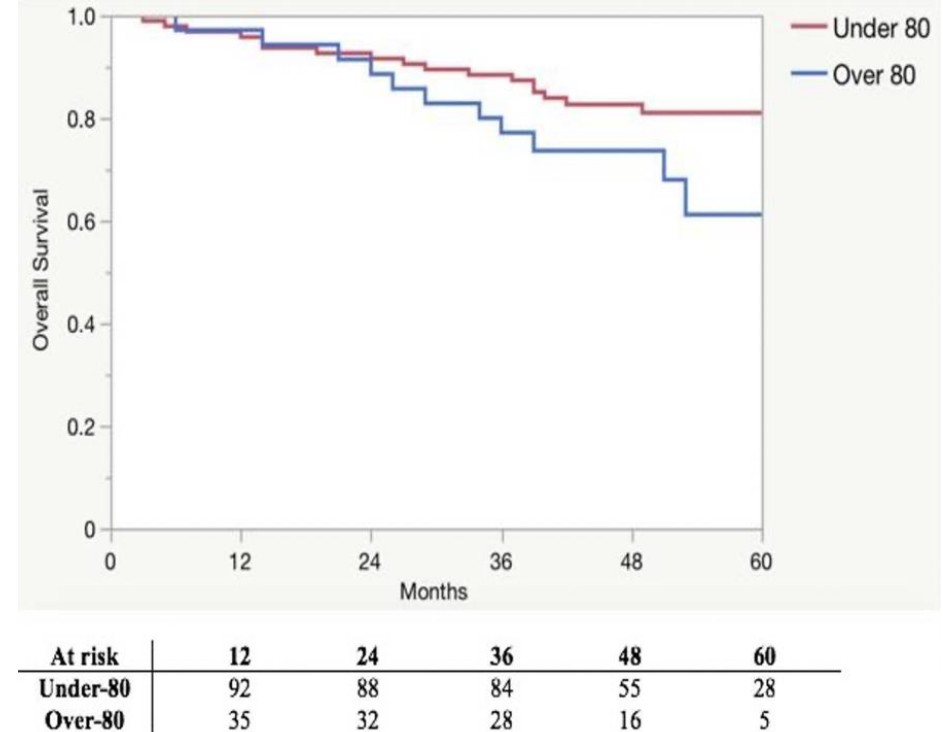

**Figure 1.** Kaplan–Meier plots of 5-year overall survival.

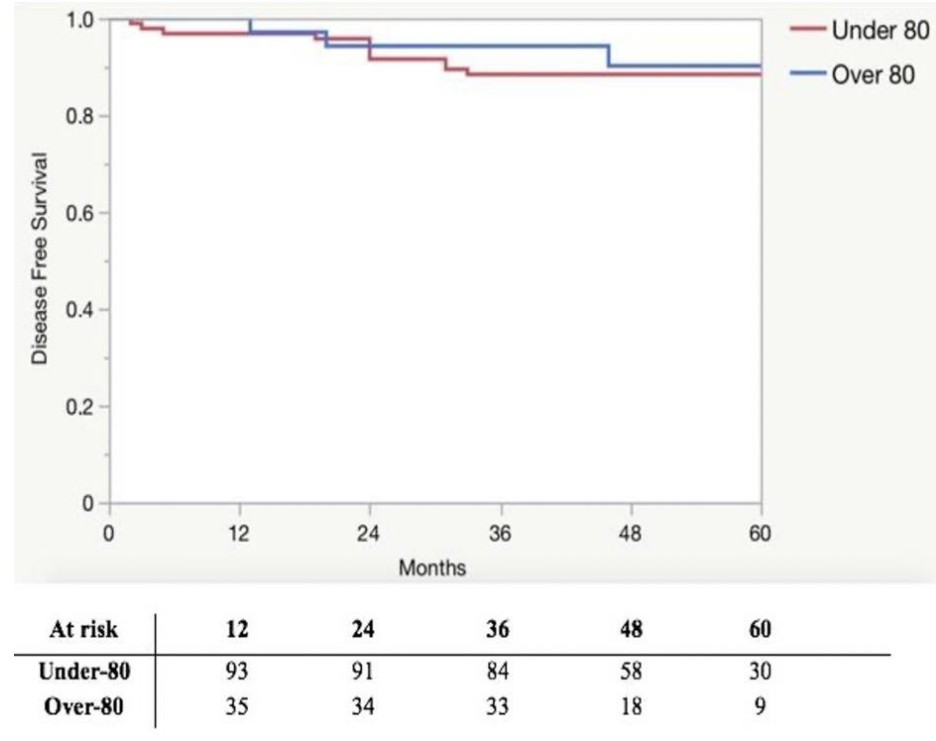

**Figure 2.** Kaplan–Meier plots of 5-year disease-free survival.

In the univariate analysis, CCI > 6 and ASA score > 2 were detected as possible risk factors for overall complications ($p < 0.05$). In the multivariate analysis, only an ASA score > 2 (OR 3.11, CI 1.3–7.6, $p = 0.01$) was an independent predictor of overall complications. Univariate and multivariate analyses for overall complications in the overall cohort are reported in Table 4.

**Table 4.** Univariable and multivariable analyses for overall complication.

| | Univariate Analysis | | Multivariable Analysis | |
|---|---|---|---|---|
| | OR (95% CI) | *p*-Value | OR (95% CI) | *p*-Value |
| Age $\geq$ 80 | 1.327 (0.553–3.182) | 0.525 | 1.146 (0.432–2.893) | 0.777 |
| Sex (Male) | 1.082 (0.486–2.410) | 0.845 | 1.364 (0.577–3.269) | 0.478 |
| ASA > 2 | 3.059 (1.344–6.960) | 0.006 | 3.114 (1.314–7.630) | 0.009 |
| BMI > 30 | 1.244 (0.466–3.321) | 0.662 | 1.081 (0.354–2.500) | 0.885 |
| CCI > 6 | 5.125 (2.134–12.304) | 0 | | |
| Tumor Site (Hepatic Flexure/Trar | 0.769 (0.262–2.262) | 0.633 | 0.834 (0.241–2.500) | 0.755 |
| Tumor pStage > 2 | 1.570 (0.647–3.806) | 0.315 | 1.382 (0.518–3.519) | 0.507 |
| Conversion to open surgery | 1.55 (0.13–17.66) | 0.723 | | |

Abbreviation: OR: odds ratio, CI: confidence interval, ASA: American Society of Anesthesiologists, BMI: body mass index, CCI'' Charlson comorbidity index, pStage: pathological Stage.

## 4. Discussion

In this study, we evaluated the effect of age on postoperative outcomes among patients undergoing laparoscopic right colectomy with CME for RCC showing that the procedure was safe and offered similar oncological outcomes, even in elderly patients.

The relevant social changes, which have occurred over the recent decades have led to a demographic transformation with an increase in the older adult population. Colon cancer is diagnosed at a median age of 69 years with 36% of newly diagnosed patients having 75 years or more [22]. Consequently, the number of elderly patients eligible for colon cancer surgery is progressively increasing; therefore, adequate treatment is required for these patients in terms of both surgical safety and oncological radicality.

Embryologic-guided resection along avascular planes was first adopted in the treatment of rectal cancer and resulted in better oncologic outcomes, especially in significant reduction of local recurrences [23]. When this concept was applied to colonic cancer resection, it consisted of dissection along the so-called 'mesocolic plane' and was defined as CME [1]. CME was associated with both a greater amount of mesocolic tissue being removed and a greater degree of lymph nodal clearance, compared to traditional colonic resection; thus, it was suggested as an approach that could potentially improve patient survival [24]. Accordingly, since 2011, the National Comprehensive Cancer Network has recommended CME as a standard procedure in the treatment of locally advanced colon cancer (i.e., T3-4, N positive or circumferential margin threatened/affected) [25,26]. Nevertheless, although emerging evidence has demonstrated an improvement in long-term outcomes without affecting patient safety, CME has not gained wide diffusion worldwide, and many colorectal surgeons have been reluctant to endorse it, mainly due to both technical and oncological concerns [18].

Relevant doubts have been raised in the execution of CME, especially in elderly patients considering, on one hand, the greatest burden of this approach in terms of excised tissue and vascular dissection compared to traditional approaches and, on the other hand, the supposed greatest frailty of this group of patients, due to their physiological degeneration and reduced functional reserve [6]. However, this controversial topic has remained under-investigated, with a lack of studies specifically focusing on it. In the present study, we sought to evaluate the safety of laparoscopic CME for the treatment of RCC in elderly

patients. With respect to a recent study from China, which compared outcomes between elderly and middle-aged patients undergoing laparoscopic CME by dividing the groups based on an age of 70 years, the present study used a higher age threshold, according to the demographic characteristics of Western populations, and elderly patients were considered those with an age $\geq 80$ years.

The impact of age in patients undergoing a laparoscopic right colectomy was analyzed and it was demonstrated that it did not significantly jeopardize postoperative outcomes [27]. Denet et al. reported the need for intraoperative blood transfusions, yet not the patient's age, as the only independent risk factor of postoperative complications [27]. Perioperative blood transfusion is an already well-known factor in potentially increasing pre- and postoperative morbidity and mortality [28]. Some authors claimed that CME could be associated with increased intraoperative blood loss [29]; however, several studies demonstrated that CME did not correlate with higher bleeding and transfusion requirement [16,30,31]. In the present series, intraoperative and postoperative transfusion rates were similar in both groups; moreover, no significant differences in postoperative surgical outcomes were found between the patients under 80 and over 80, with similar rates of overall complications, major complications, and 90-day mortality rates between the groups. Consistently, only an ASA score > 2 was an independent predictor of overall complications at multivariate analysis. The ASA score represents a preoperative clinical condition resulting from several pathophysiological factors, and its ability to predict postoperative complications was widely demonstrated [32]. These findings suggest that concomitant pathologies and general conditions could be the main factors affecting postoperative outcomes after laparoscopic right colectomy with CME, and thus, it should not be denied to patients on the basis of age alone. Differently, median LOS was slightly higher in the over-80 group (8 days vs 5 days, $p < 0.001$), which could be due to lower adherence to the ERAS pathway among elderly patients, the need for a seamlessly organized and well-coordinated transition from the hospital to the domestic life, or a more frequent need for post-discharge institutionalization. In a recently published retrospective study, which investigated predictive variables of ERAS discontinuation in patients undergoing colorectal surgery, an age $\geq 75$ years was independently associated with ERAS failure [32]; however, this result has not been unanimously confirmed in other surgical fields, namely in gastric surgery, where frailty and comorbidity, yet not age, per se, represented the main limit for the completion of an ERAS program [33]. Consistently, recent studies have demonstrated that while frailty is strictly related to age, it should be considered a multidimensional condition affecting not only elderly patients but young people too, thereby recognizing the potential difference between chronological and physiological age [34]. In this scenario, independently from the patient's age, the introduction of a multimodal prehabilitation program could be a useful and cost-effective tool to improve postoperative outcomes including LOS, in both young and elderly patients [35,36].

Many studies have suggested potential oncological advantages of CME in reducing local recurrence and increasing the survival rate of colon cancer patients [3,4,37]. After the introduction of CME, the 5-year local recurrence and cancer-related survival rates in stage III colon cancer patients were reported to shift from 14.8% to 4.1% and from 61.7% to 80.9%, respectively [4]. In particular, in a population of elderly patients ($\geq$70 years) undergoing laparoscopic right colectomy with CME, the 5-year OS, and 5-year DFS rates were 71% and 55%, respectively [6]. In the present study, we demonstrated a 5-year OS and 5-year DFS rate of 61.2% and 90.2%, respectively, in patients aged $\geq 80$ years. Interestingly, although a slight difference was found between the groups, in terms of OS, which could be considered an expected result since the likelihood of death for any reason inherently increases with age progression, meaning an overall shorter life expectancy for elderly people, no statistical difference was found between the under-80 and over-80 groups relating to oncological outcomes. As highlighted by large retrospective studies, elderly patients with colorectal cancer are less likely to receive adjuvant chemotherapy, and the rate of patients who received adjuvant treatment declined dramatically with chronologic age, ranging from

78% for patients aged 65–69 years to 11% in those > 85 years [38,39]. In the present series, in the over-80 group, only 1 patient out of 6 was able to receive adjuvant chemotherapy, with a relevant difference compared to the under-80 group (16.7% vs. 96.1%, $p < 0.0001$); this low rate of adjuvant chemotherapy performance was consistent with the literature and was not due to postoperative complications, which were comparable between the groups. Notably, the over-80 group showed a DFS at 12, 36, and 60 months of 100.0%, 94.3%, and 90.2%, respectively, without a difference in the under-80 group, where it was 96.8%, 88.4%, and 88.4% ($p = 0.536$), respectively, suggesting a potential protective effect of CME in patients who did not receive adjuvant chemotherapy for any reason. Thus, CME could play a pivotal role in the therapy of CRC, especially in elderly patients who can rarely benefit from adjuvant chemotherapy due to their frailty and comorbidity, ensuring adequate DFS without increasing postoperative surgical complications.

The current study has several limitations. Even if prospectively collected, the data were retrospectively analyzed and were based on a single-center experience, potentially limiting their validity in other settings. In addition, surgical techniques for laparoscopic right colectomy with CME may differ among centers; thus, hindering the comparison of the results among studies. The relatively low number of "events" in the OS and DFS calculations prevented us from performing a proper multivariable Cox regression model. Additionally, the small number of considered patients, especially in the over-80 group, may have led to the generation of a type-II error during the comparison of the two groups, which has prevented a strong conclusion from being reached. Given the retrospective nature of the study, a bigger number of patients would raise the power of the statistical testing. Therefore, in our case, a multicenter study would be desirable to raise the number of included patients. The scarce data of the literature makes it difficult to compare our data with the experiences of others. Even if recurrence in colon cancer mostly occurs within 2 years after surgery [40], the shorter median follow-up of the elderly group (32.5 vs. 43 months) might have affected the long-term results.

## 5. Conclusions

In conclusion, laparoscopic right colectomy with CME for RCC was safely performed in elderly patients ensuring similar oncological outcomes irrespective of age. In the current aging society, surgeons are increasingly treating more elderly patients with cancer. Elderly patients with RCC should not be precluded from appropriate resection with CME solely due to age, even when considering the scarce chance of receiving adjuvant chemotherapy.

**Author Contributions:** All authors contributed to the conception and design of the study. Material preparation, data collection, and analysis were performed by L.R., A.G. and M.M. The first draft of the manuscript was written by L.R., A.G. and M.M. and all authors commented on previous versions of the manuscript. M.M., L.R., A.G., P.C., M.O., B.A., I.G., P.A., C.L.B., C.M. and G.F. All authors have read and agreed to the published version of the manuscript.

**Funding:** This research received no external funding.

**Institutional Review Board Statement:** All procedures performed in studies involving human participants were in accordance with the ethical standards of ASST Grande Ospedale Metropolitano Niguardaand with the 1964 Helsinki Declaration and its later amendments or comparable ethical standards.

**Informed Consent Statement:** For this type of study formal consent is not required.

**Data Availability Statement:** The data presented in this study is available on request from the corresponding author.

**Conflicts of Interest:** The authors declare no conflict of interest.

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
