# Peer review of "Should Laparoscopic Complete Mesocolic Excision Be Offered to Elderly Patients to Treat Right-Sided Colon Cancer?"

_curroncol, doi:10.3390/curroncol30050376_

Round 1

Reviewer 1 Report

I have considered the manuscript by Michele Mazzola on Should laparoscopic Complete Mesocolic Excision be offered to 2 elderly patients for the treatment of right colon cancer?

This is a well written manuscript and the topic is interesting.  My main concerns are:

1. The sample size is very small to make strong conclusion as only 35 patients in the elderly group.

2. It is unclear that CME was performed as there are no operative details (nor postoperative confirmation of the dimensions of the specimens; photographs to show CVL, apart from the number of the harvested LNs, which is modest for CME. can the authors provide these information to reassure the readers that CME was performed in all cases. 

Author Response

We thank the Reviewer for the comments.

1 As observed by the Reviewer the small sample size should be highlithed as a limit of thi study. Accordingly, a sentence has been added in tha last part of the discussion section.

2 We fully understand the comment of the Reviewer. Unfortunately, due to the retrospective nature of the study, the data required about the specimen features are not available. However our technique for CME is based on the vascular anatomy of the patient as a roadmap for dissection. Our technique was widely described in several previous papers that we cited in the present study. According to the observation of the Reviewer, we added a citation (number 17) where our technique was decribed more deeply and explained with a video.

Reviewer 2 Report

1. Please explain how you selected the timeframe of your analysis (2015-2018). CME technique was described almost a decade before the time you started analysis. 

2. Please provide more information on how lap right colectomy with CME was performed at your center, how many surgeons use this technique and how may cases they do annually. 

3. What were the criteria to allocate CME in your patients? This is important to inform readers on potential confounders. 

4. You report on the median follow up for the entire cohort; please provide the median and 95% CI of follow up for each of the two patient groups.  

5. OS is not an accurate surrogate of disease specific survival especially in older patients. Do you have data on DSS? If not, you have to acknowledge this limitation of your study.

6. You need to adjust for confounders in a multivariable Cox regression model; log ranks tests do not suffice. Please add that analysis for the entire cohort. 

Author Response

1. In our department, CME was implemented as the standard surgical approach for RCC from September 2015. 

2. All patients were operated on by colorectal surgeons having at least 50 laparoscopic right colectomies with CME completed at the time of the analysis
(which is suggested to be the number required to acquire proficiency) were considered. CME has been adopted at our division for all RCCs since September 2015 and all the surgeons considered began to apply it simultaneously from that date. The three abovementioned surgeons all had some degree of experience in minimally invasive colorectal surgery and were part of a dedicated surgical team that had obtained specialized training in colorectal surgery. In particular, the first surgeon , before starting to perform CME, had done 34 laparoscopic colorectal resections, while the second surgeon had performed 110 and the third surgeon 95. All three had already carried out at least 50 conventional laparoscopic right colectomies (110, 55 and 50 cases, respectively).

3 Due to the poor ability of preoperative staging exams to discriminate patients affected by T1 N0 neoplasms, we sistematically adopted CME for all patients affected by any primary adenocarcinoma of the cecum, the ascending colon, the hepatic flexure, and the first-third of the transverse colon.

4. As suggested, the median FU for the two cohorts was added: "52 (IQR 44.5-62) in the under-80 group and 50 (IQR 39-58) months in the over-80 group (p=0.242).

5. We thank the reviewer for the comment. In the study we reported both overall survival and disease free survival as secondary endpoints, reflecting two important aspects to take into consideration. OS is influenced both by the specific disease that we considered (colon cancer) and by other causes that are otherwise of critical importance when analysing the impact of surgery in elderly patients: it tells us if surgery for cancer is a valid option for patients who have de facto a reduced life expectancy for other than oncological reasons. DFS illustrates if there are differences in terms of recurrence in the two cohorts, and thus if cancer has different natural history in elderly patients, who additionally have less chance to access adjuvant treatments.

6. The reviewer comment is certainly pertinent. Unfortunately, the relatively low number of "events" (e.g. death at 5 years in the case of OS) did not allow to run an adequate and significant multivariable Cox regression model. Additionally, OS and DFS were secondary endpoints. Multivariable analysis was reported for overall complication instead. Otherwise, as suggested, we reported this issue in the limitation paragraph: The relatively low number of “events” in the OS and DFS calculation prevented us from running a proper multivariable Cox regression model.  

Reviewer 3 Report

The choice to propose CME in all patients with right colon cancer without regard for the preoperative staging is questionable and should be expressly justified in the paper. The distribution among stadies is uncommon: 48% of older patients are in stage I in the over 80 group. In this population only 6 patients have an indication for adjuvant chemotherapy, therefore is impossible to draw any conclusion based on this very small population. 

Author Response

1 We agree with the Reviewer that the choice to perform CME in all patients affected by colonic cancer may appear questionable; however it is a widely adopted practice, especially in Western countries, due to the low accuracy of preoperative staging in discriminating between stage I and stage II disease. In addition, recent evidence, reported that CME is associated with significantly impreoved DFS, compared with conventional colon resections, in particular for patients with UICC stage I or II disease, without jeopardizing surgical outcomes or increasing morbidity. According to the Reviewer's suggestion we have added a sentence in the method section to better explain our choice.

2 we fully agree with the Reviewer and consequently we have added a sentence in the limit paragraph, highlighting that due to the small sample size strong concluion cannot be drawn

Round 2

Reviewer 2 Report

Thanks for revising according to my comments.